# Enhancing Human Biomonitoring Studies through Linkage to Administrative Registers–Status in Europe

**DOI:** 10.3390/ijerph19095678

**Published:** 2022-05-06

**Authors:** Helle Margrete Meltzer, Tina Kold Jensen, Ondřej Májek, Hanns Moshammer, Maria Wennberg, Agneta Åkesson, Hanna Tolonen

**Affiliations:** 1Division of Climate and Environment, Norwegian Institute of Public Health, 0213 Oslo, Norway; 2Department of Clinical Pharmacology, Pharmacy and Environmental Medicine, University of Southern Denmark, 5230 Odense, Denmark; tkjensen@health.sdu.dk; 3Institute of Health and Information and Statistics of the Czech Republic, 12801 Prague, Czech Republic; ondrej.majek@uzis.cz; 4Faculty of Science, RECETOX, Masaryk University, 62500 Brno, Czech Republic; 5Institute of Biostatistics and Analyses, Faculty of Medicine, Masaryk University, 62500 Brno, Czech Republic; 6Department of Environmental Health, Center for Public Health, Medical University Vienna, 1090 Wien, Austria; hanns.moshammer@meduniwien.ac.at; 7Department of Hygiene, Medical University of Karakalpakstan, Nukus 230100, Uzbekistan; 8Section of Sustainable Health, Department of Public Health and Clinical Medicine, Umeå University, 90187 Umeå, Sweden; maria.wennberg@umu.se; 9Institute of Environmental Medicine, Karolinska Institutet, 17177 Stockholm, Sweden; agneta.akesson@ki.se; 10Department of Public Health and Welfare, Finnish Institute for Health and Welfare, 00300 Helsinki, Finland; hanna.tolonen@thl.fi

**Keywords:** biomonitoring, health registers, administrative registers, HBM4EU, health, chemicals

## Abstract

Record linkage of human biomonitoring (HBM) survey data with administrative register data can be used to enhance available datasets and complement the possible shortcomings of both data sources. Through record linkage, valuable information on medical history (diagnosed diseases, medication use, etc.) and follow-up information on health and vital status for established cohorts can be obtained. In this study, we investigated the availability of health registers in different EU Member States and EEA countries and assessed whether they could be linked to HBM studies. We found that the availability of administrative health registers varied substantially between European countries as well as the availability of unique personal identifiers that would facilitate record linkage. General protocols for record linkage were similar in all countries with ethical and data protections approval, informed consent, approval by administrative register owner, and linkage conducted by the register owner. Record linkage enabled cross-sectional survey data to be used as cohort study data with available follow-up and health endpoints. This can be used for extensive exposure-health effect association analysis. Our study showed that this is possible for many, but not all European countries.

## 1. Introduction

Human biomonitoring (HBM) aims to assess nutritional status and/or the exposure to environmental substances or their metabolites in the human body, usually through analyses of blood, urine, hair, breast milk, or tissues in individuals or a population group. This allows for measurement of cumulative exposure to substances by all possible pathways, i.e., inhalation, digestion, and direct contact [1]. Cross-sectional biomonitoring data can be used to obtain insight into the range and determinants of various exposures, including the relative relevance of various exposure pathways—especially if combined with other types of data. However, individual exposure data, accompanied by additional personal information, enables new research, such as exploring the potential health consequences associated with the exposures under study. Answering that type of research questions has traditionally required a prospective cohort-study design where participants are followed over time for the investigation of later health outcomes. 

In theory, such follow-up can be performed through repeated surveys. However, as for any survey, HBM surveys require a high participation rate in order to be representative of the target population and, more importantly, to not introduce bias of the health outcomes of interest due to loss at follow-up. This is a laborious and costly approach with high likelihood of declining participation rates as the study proceeds because of high participant burden and the risk of the organizers losing contact information [2].

An alternative to survey questionnaires and self-reported health information from follow-up studies is to obtain this information from administrative health registers, often also called routine data or real-world data, such as causes of death registers, hospitalization registers, cancer registers, and/or prescription medication registers. Emerging new sources of data include administrative registries or electronic health records [3]. Record linkage is increasingly being used to combine electronic records containing information from different sources about an individual, organization, or location. Linkage offers a relatively quick and low cost means of capturing information from large administrative datasets for service planning, delivery and evaluation, surveys and censuses, and research [3]. Some countries already have extensive experience with linking diverse data sources [4,5,6,7], while other countries are beginners or cannot perform such linkages because of very strict data protection laws.

A comprehensive overview of which countries allow for merging of data has been lacking. In this study, we have investigated the availability of health registers in different EU Member States, EEA countries, and Israel and have assessed whether they can be linked to HBM studies. 

## 2. Methods

### 2.1. Setting

Our starting point is the European Joint Program “HBM4EU”, a joint effort of 30 countries and the European Environment Agency, co-funded under the European Commission’s Horizon 2020 program, grant agreement No. 733032, for advancing and implementing HBM on a European scale and for providing scientific evidence for chemical policy making [8,9]. The present specific data linkage project was embedded within a Work package on Linking HBM studies, health studies, and health registers. 

### 2.2. Questionnaire

Within the framework of the HBM4EU project, an online questionnaire about the availability of health-related, national, or large regional, administrative registers and the possibility to link them to the HBM studies was prepared [10]. The questionnaire included questions about country, the official name of the register, type of register, if they were disease-specific, if they included a unique person identifier, conditions for access to data, the coverage of the register, problems with representativity, last update year, update frequency, what type of information the register contains, and technical questions related to availability, timing, and costs. 

Within the HBM4EU project, each country established a National Hub consisting of the relevant stakeholders working on the collection and use of HBM data for policy making and research. A questionnaire link was distributed to the National Hub coordinators, or a country representative with the relevant knowledge, three times: in 2017, 2019, and at the end of 2020–early 2021. In each round, national hub coordinators were asked to add possible missing information. Information from new countries was obtained or previously provided information was updated (2017: 19 countries; 2019: 22 countries; 2020: 28 countries). 

In addition to the questionnaire, more detailed information was obtained from six countries (Austria, Czech Republic, Denmark, Finland, Norway, and Sweden) on how record linkage between biomonitoring survey data and administrative register data could be done. These countries were selected based on their participation in the HBM4EU project and experience on record linkage.

When comparing the results provided by the National Hub coordinators with other published data on availability of administrative health-related registers, it was apparent that some information was incomplete. Therefore, we also used additional information sources such as European Health Information Portal [11], European Network of Cancer Registries [12], Eurostat website on mortality [13], OECD Health Statistics 2021 Definitions, Sources and Methods [14], FamilySearch Research Wiki [15], and an EU report on electronic health reports and ePrescriptions [16]. 

## 3. Results

### 3.1. The Questionnaire

Results are based on responses obtained from the HBM4EU National Hub coordinators or the country representatives identified to have the best knowledge of their administrative health registers. Responses were received from 28 countries (Figure 1). 

### 3.2. Availability of Different Registers

The evaluation questionnaire covered only nationally representative administrative registers, i.e., regional registers were not included unless there were several of them that together covered the entire country. The availability of nationally representative administrative health registers varied substantially between countries. Vital registrations including births and deaths were available either at the national or regional level in all countries responding to our questionnaire (Figure 2). Cancer registers were one of the disease specific registers with long history in Europe and, therefore, all countries had either national or several regional cancer registers covering the whole country (Figure 3).

Registers for hospitalizations (in- and/or out-patient) were less frequently reported by the National Hub coordinators. Only 50% reported that their country had a register for hospitalizations (Figure 4). In many countries, electronic health records were collected and combined at the national level [16], but it was not clear whether these could also be seen as registers for hospitalization and, thus, used for record linkage. For example, in Austria, health insurance companies possess a huge amount of individual health data, but they are not willing to share these data for research purposes.

A similar situation was found for medical prescription registers, i.e., only 11 countries reported that they have this type of register (Figure 4). Many countries had already implemented ePrescriptions systems [16], but the use of ePresciptions varied between countries and in some countries, they were used only by hospitals, not by all health care providers. Therefore, it was not clear if these ePrescriptions systems could be considered as a medical prescription register. 

In a number of countries, other disease specific registers were also available, and registers exist for injuries and accidents, diabetes, myocardial infractions, stroke, rare diseases, tuberculosis, and HIV/AIDS.

### 3.3. Availability of Personal Identifiers

For record linkage of any two data sources, having identifiers that allow for one-on-one matching of data on the level of individuals is required. Ideally, one unique identifier is available in all linked data sources, but linkage can also be done using several identifiers if necessary. Based on the information received through our questionnaire, in EU member states/EEA countries, national personal identifiers are common (25/28 countries), but they are not always used systematically in all data sources such as administrative registers. In half of the countries (14/28), a national identification code was used systematically in all of their administrative registers, and additionally two countries (Lithuania, and Latvia) used a national identification number in most of the registers (Figure 5).

Spain has a national identification number in the identification document, which is issued to persons after age of 14 years, i.e., children are not covered. Switzerland has a social security number, which is used in some of registers, and in Greece, the identification number changes every time a person is issued with new identification card. Austria has two personal identification numbers, with one unique social security number that is known to the individual and could potentially be used for record linkage. The Austrian statistics institute (Statistik Austria, StatAT) has developed its own personal identifier. This allows StatAT to cross-link its own registers, e.g., cancer to cause of death or to hospital admission, but a researcher has no way of getting access to that ID. In countries such as Greece, Israel, Spain, Switzerland, and Portugal, record linkage is either restricted or not allowed at all by national legislation. 

### 3.4. Examples of Record Linkage Procedures

There are two main ways of conducting record linkages [17]. If the same personal identifier is available in both the HBM survey data and in the linked administrative register, **deterministic record linkage** can be performed. If unique identifiers are missing, then **probabilistic record linkage** can be performed. This approach uses a number of identifiers in combination to identify and evaluate links. 

The actual process and protocol for record linkage varies between countries, due to national legislation, but some basic ethical principles and data protection requirements applied to all. This means that, in most countries, ethical approval and possibly also approval from the data protection authority was required. When survey data are linked to the administrative data sources, most countries also require that survey participants provide an informed consent for the record linkage. In addition, approval/permission from the register owner/controller is needed before data can be obtained.

We studied details for the processes of record linkage in six European countries (Austria, Czech Republic, Denmark, Finland, Norway, and Sweden) linking HBM survey data to the health-related administrative resisters such as birth register, hospitalizations, cancer register, or mortality register. Table 1 provides an overview of the required procedures in each country.

In all countries, ethical approval is required for HBM survey protocol, which also means that ethical approval is required for record linkage. Practices on data protection approval and need for informed consent vary between countries. In all cases, permission by register owner/controller is required prior to the record linkage and linkage is done by the register owner. In most cases, the national personal identifier exists in all data sources allowing deterministic record linkage.

In Austria, record linkage for scientific purposes is allowed and informed consent is not necessary. In practice, this can only be done with the mortality data from the causes of death register and the procedures include obtaining an approval from an ethics committee.

## 4. Discussion

Record linkage between HBM survey data and data from administrative registers provides researchers with a more extensive data source. Record linkage can help overcome data problems such as missing information on socio-demographic variables. It can also provide reliable information on diagnosed diseases and use of medications for follow-up of health and vital status of the established cohorts.

Through record linkage, we can reduce participant burden, reduce the cost of data collection, reduce the survey measurement error in some cases, and increase the amount of information available for each individual, avoid recall bias, which is often present in self-reported data, and adjust our analysis for wider range of potential confounders [18,19,20,21,22].

HBM survey data together with linked administrative data have been used, for example, to study cancer incidence among exposed [23,24,25], association between exposure and onset of disease other than cancer [26,27,28], and mortality follow-up of cohort participants [29,30]. 

Our study showed that the availability of administrative health registers varies substantially between the EU Member States and EEA countries. Birth registers, registers for causes of death, and cancer registers are available in all countries at least at the regional level. Registers of hospitalizations and medical prescriptions are not as common, but available in some countries. Unfortunately, not all available registers can be used for record linkage due to lack of required identifiers or national legislation, and it is usually data protection regulations related to sensitive health data preventing that. 

However, both health and administrative registries are usually generated for other purposes than research. Data may accumulate over a long time covering the entire population. This has both pros and cons for the data as such. Since data accumulate as part of the ongoing administrative process, such as recording of hospital visits or mortality, the cost of data generation is low and usually the coverage of the target population is good. On the other hand, routine registration is performed by a large number of people who are not always using standardized operating/recording procedures, which may hamper data quality, and the coding systems may evolve over time. In addition, administrative databases cover only topics relevant for their original purpose (e.g., causes of death, reasons for hospital visit), which may limit available/desired information.

The importance of administrative health data for research has been recognized also at the EU level. The European Commission (EC) has launched the creation of European Health Data Space (EHDS) as one of its priorities for 2019–2035. EHDS aims to promote the secondary use of health data through better exchange and access to health data, including electronic health records, genomics data, data from patient registers, etc. [11]. This should enhance the availability, quality, and interoperability of health data across the EU.

Technically, it is relatively easy to perform record linkage if both survey data and the administrative register have the same unique personal identifiers, such as a national identification code. In the European Union and EEA countries, this seems to be the case in approximately 50% of the countries. In addition to this, in some countries, common identifiers are used only for some of the administrative registers. When common identifiers are not available, record linkage can still be done using probabilistic methods, but this is much more burdensome.

The quality of record linkage is strongly dependent on the availability of the required identifiers in the linked datasets. Deterministic linkage is generally high quality and fast, if the number of missing identifiers is minimal and their accuracy is high. Furthermore, probabilistic linkage can be of high quality when the identifiers that are used have high coverage in all linked datasets [31,32,33].

To conduct a record linkage generally requires informed consent from the survey participants. Obtaining the informed consent may be challenging and in case of selective non-consent, cause bias to the outcomes. The consent rate has been shown to vary between 19–97% [34]. Some studies have found that consent for record linkage of survey data is associated with sociodemographic characteristics and health status, but results are not consistent across studies. This may be due to the differences in the target population of the studies (i.e., patients vs. general population vs. different age groups) as well as administrative registers used for record linkage [20]. 

Conducting a record linkage between an HBM or any other survey and administrative registers should be incorporated in the planning of the survey since most countries require that record linkage is described in the study plan, information about the linkage is provided for the survey invitees, and participants provide an informed consent for the linkage. For most countries, record linkage also needs to be described in detail for the ethical approval process, that is, to what registers the linkage will be done. It is essential to check the national regulations and guidelines in advance to ensure that all required steps will be taken in right order and all required documents are prepared. 

Our more in-depth investigation of record linkage procedures in six countries demonstrated that procedures have similarities but still differed between countries. While the four Nordic countries Denmark, Finland, Norway and Sweden have decades of experience with this type of linkage, and the process is fairly smooth, the same exercise may be challenging in other countries, partly because ethical committees and register owners are not familiar with this type of data use. 

HBM studies are costly and are therefore often conducted in small samples with too low statistical power to study rare health outcomes. Thus, studies should be combined, e.g., across countries, to obtain a large enough sample size. This will require the availability of similar administrative registers in all involved countries, and that data linkage is possible, i.e., the required identifiers exist. This has been done in Nordic collaborations [21,35,36] with similar registers and procedures for collecting information. Despite these similarities, issues about data quality need to be considered [21]. 

The strength of our evaluation is that we have used several data sources to assess the availability of different registers in different countries, not relying only on self-reported information through the questionnaire. We have also added an in-depth analysis of six countries on the procedures of record linkage. A limitation with our evaluation is that it covers only countries that were partners in the HBM4EU project and that responded to the questionnaire, and in some cases, the respondent may not have had the comprehensive knowledge on the availability of health-related administrative registers in their country and their possibilities for record linkage.

## 5. Conclusions

Record linkage is a cost-effective way to obtain comprehensive morbidity and mortality follow-up and medical history for survey participants. This allows us to transform originally cross-sectional HBM studies into cohort studies with follow-up, which can be used for extensive exposure-health effect association analysis. At the European level, possibilities for record linkage vary substantially between countries. Therefore, using record linkage to obtain information for cross-country combined HBM studies is challenging and, in some countries, not currently possible. 

The COVID-19 pandemic has demonstrated the need and power of data obtained from different data sources, including health registers, and combining them for effective information generation to support policy decisions. This experience should be transferred to other fields of research including environmental health research. 

## Figures and Tables

**Figure 1 ijerph-19-05678-f001:**
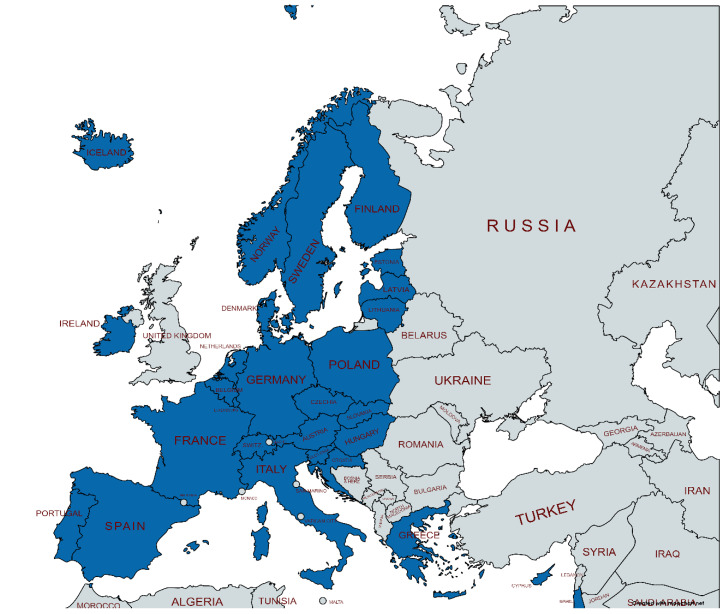
Contributing countries. Figure has been created using MapChart.net web tool.

**Figure 2 ijerph-19-05678-f002:**
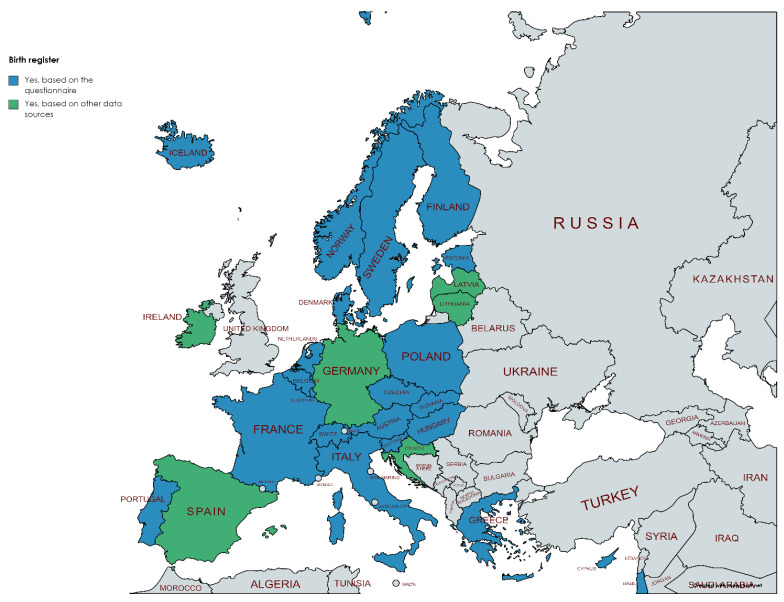
Availability of civil registers for births and deaths (blue = provided by the National Hub contact points, green = obtained from other data sources as described in the Section 2). Figure has been created using MapChart.net web tool.

**Figure 3 ijerph-19-05678-f003:**
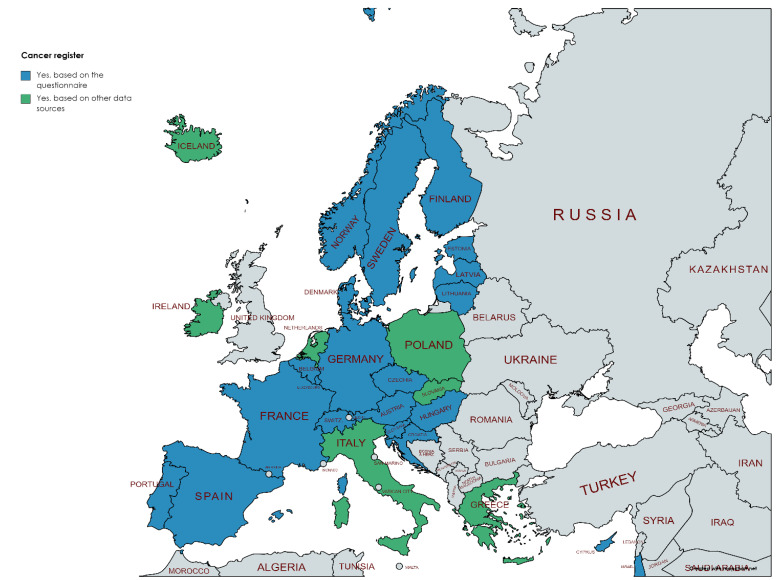
Availability of cancer registers (blue = provided by the National Hub contact points, green = obtained from other data sources as described in the Section 2). Figure has been created using MapChart.net web tool.

**Figure 4 ijerph-19-05678-f004:**
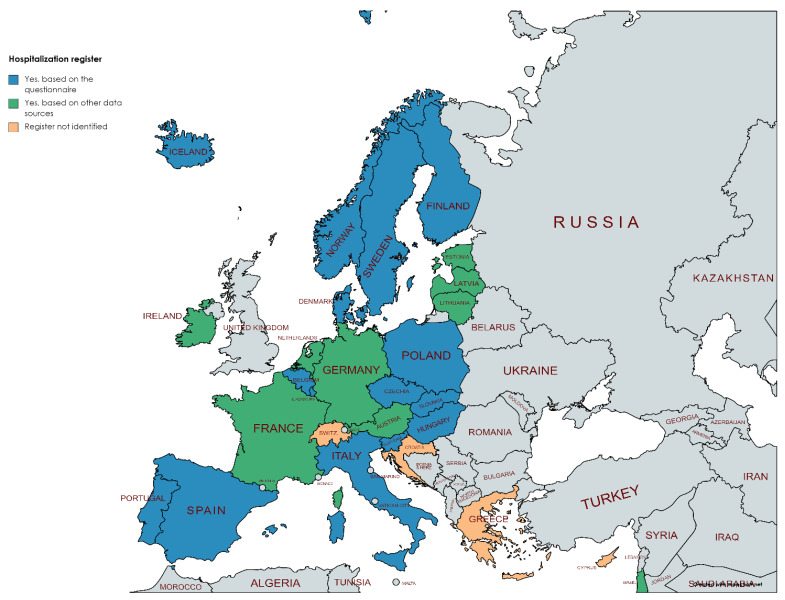
Availability of register of hospitalizations and medical prescriptions (blue = provided by the National Hub contact points, green = obtained from other data sources as described in the Section 2). Figure has been created using MapChart.net web tool.

**Figure 5 ijerph-19-05678-f005:**
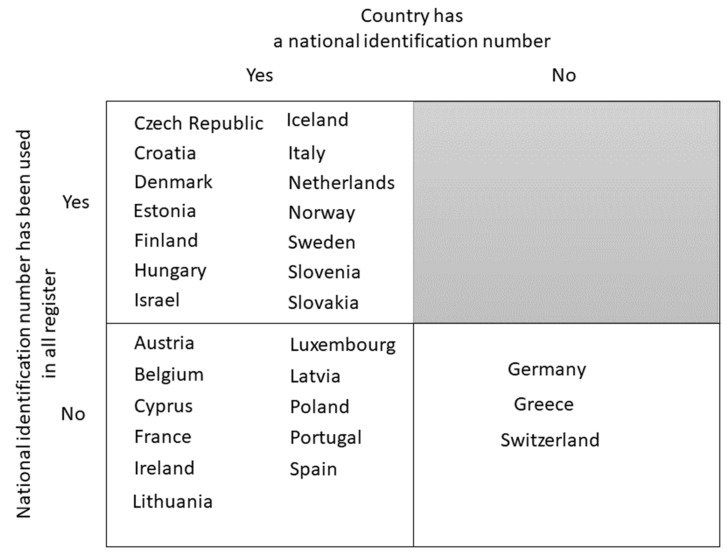
Availability of national identification number and its use in different registers.

**Table 1 ijerph-19-05678-t001:** Example of procedures for record linkage in six countries.

	Ethical Approval Including Research Plan	Approval from Data Protection Authority	Informed Consent	Permission by Register Owner/Controller	Identifier(s) Used for the Record Linkage	Methods of Data Linkage	To Which Registers HBM Data Can Be Linked	Who Performs the Record Linkage
Austria	Required	Not required	Not required *	Required	Name and date of birth	Probabilistic	Mortality and cancer registers. Currently not possible to link to hospitalizations/patient data due to concerns over privacy issues	The register owner
Czech Republic	Required	Not required	Required	Required	National personal identification number	Deterministic	National health registers, potentially including reproductive health, hospitalizations, cancer, mortality, and cardiovascular surgery	The register owner
Denmark	Required	Required	Required	Required	National personal identification number	Deterministic	Both health-related and non-health related registers	The register owner
Finland	Required	Not required, evaluated as part of ethical approval process	Required	Required	National personal identification number	Deterministic	All health-related registers including birth, mortality, hospitalizations, medical prescriptions, cancer, malformation, infectious diseases, and vaccinations Several non-health registers such as marital status, education, and sociodemographic position	The register owner
Norway	Required	Required	Required	Required	National personal identification number	Deterministic	Health-related registers including birth, hospitalizations, cancer, medical prescriptions, and mortality	The registry owner
Sweden	Required	Required	Required	Required	National personal identification number	Deterministic	Both health-related and non-health related registers	The register owner

*: Informed consent: not necessary in general. But since an ethical approval is needed, it can be that the ethics committee asks for it.

## Data Availability

All data used in this paper have been referred to in our methods section or in our reference list.

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
