# Peer review of "Enhancing Human Biomonitoring Studies through Linkage to Administrative Registers–Status in Europe"

_ijerph, 2022, doi:10.3390/ijerph19095678_

Round 1
Reviewer 1 Report
There are multiple minor grammar mistakes throughout the manuscript but the following are more noticeable --
- L21: complement not compliment
- L66-67: reword this sentence as it is unclear
- L75: "...a joint effort" not "...which is a joint effort"
- L115: Remove "Presented" at the start of the sentence
- L127: change the sentence to read "...with a long history in Europe and, therefore, all countries..."
- L204-207: Reword this sentence
- L211-216: Reword this sentence
- L225: Change to "...onset of diseases other than cancer..."
- L245-251: Reword this sentence
Additionally, I have the following questions/suggestions:
- For Figures 2 through 4, it would be beneficial to color countries on the map not included as part of the study differently than those that are part of the study and do not have that specific registry
- The disease specific registries noted in L149-151 should be included as supplemental information
- Creating a single database to include all of the data included in all of the figures and tables and including it as a supplement would be useful to the scientific community in thinking about and planning studies
- L165: Two countries are listed but the text notes that it should be three countries
- The IRB, informed consent, and data availability sections are missing
Author Response
Please see attached word document.

Reviewer 2 Report
Thank you for the opportunity to review your paper. It is a good piece of work that looks more deeply into data sharing to better improve outcomes and use of HBM data that is collected in Europe.
Specific comments are as follows:
- Check the wording of the sentence starting at the end of line 163 "By half of the countries..." - it is not clear
- Table 2 is missing some letters in the headings, and text is missing from Austria in relation to 2 aspects listed
- Table 2 second last column for Czech Republic includes "etc" it would be better to include all details on the data that can be linked as it is unlikely the reader would not what else is included.
- Lines 294 to 296 - the strength outlined in this sentence is not clearly discerned from the paper. The paper outlines a range of data sources available in different countries. It is unclear where this has been used in the paper to address gaps in questionnaire results.
- Line 296 - Start of the sentence mid way on this line could perhaps be revised to "A limitation with our evaluation is that it covers..."
- It would be good to include further limitations in relation to the data linking aspect, including the type of data collected - is it comparable between different countries (e.g. are cardiovascular disease hospitalisations recorded using the same coding or approach in all the countries or does this differ) and can the data be linked to the same time period relevant to the HBM data?
